# Etiologies of Early-Onset Hearing Impairment in Rwanda

**DOI:** 10.3390/genes16030257

**Published:** 2025-02-23

**Authors:** Esther Uwibambe, Leon Mutesa, Charles Muhizi, Isaie Ncogoza, Elvis Twumasi Aboagye, Norbert Dukuze, Samuel M. Adadey, Carmen DeKock, Ambroise Wonkam

**Affiliations:** 1Center for Human Genetics, College of Medicine and Health Sciences, University of Rwanda, Kigali P.O. Box 4285, Rwanda; uwaesther04@gmail.com (E.U.); lmutesa@gmail.com (L.M.); norbertduk123@gmail.com (N.D.); 2Division of Human Genetics, Faculty of Health Sciences, University of Cape Town, Cape Town 7925, South Africa; atelvis45@gmail.com (E.T.A.); carmen.dekock@uct.ac.za (C.D.); 3Department of Ophthalmology, College of Medicine and Health Sciences, University of Rwanda, Kigali P.O. Box 4285, Rwanda; charlesmuhizi6@gmail.com; 4Department of the Ear, Nose, and Throat (ENT), College of Medicine and Health Sciences, University of Rwanda, Kigali P.O. Box 4285, Rwanda; uncogoza@gmail.com; 5National Institute on Deafness and Other Communication Disorders, National Institutes of Health, Bethesda, MD 20892-2510, USA; samuel.adadey@nih.gov; 6McKusick-Nathans Institute and Department of Genetic Medicine, Johns Hopkins University School of Medicine, Baltimore, MD 21205, USA

**Keywords:** hearing impairment, etiology, genetics, *GJB2*, *GJB6*, Rwanda

## Abstract

**Background:** Over three-quarters of the people living with hearing impairment (HI) live in low- and middle-income countries. However, Rwanda has limited data on the clinical profile of HI. **Aim:** We used community-based nationwide recruitment of participants to determine the etiology of early-onset (<7 years of age) HI in Rwanda. **Methods:** Participants were included after clinical examination, including audiological assessment by pure tone audiometry and/or auditory brainstem response. DNA was extracted from peripheral blood, and the entire coding region of *GJB2* was interrogated using Sanger sequencing. Multiplex PCR and Sanger sequencing were used to analyze the prevalence of the *GJB6*-D3S1830 deletion. **Results:** The participants were recruited from seven inclusive schools, one university teaching hospital, and four independent communities nationwide. We reviewed the clinical histories of 422 individuals affected by early-onset HI from 348 families and found that 21.18% (*n* = 89/422) was linked to early childhood meningitis infection while 51.23% (*n* = 216/422) was categorized as unknown HI etiology. Because of putative genetic causes, 82/348 (23.6%) families were reviewed and identified for genetic testing. Within the 82 families with potential genetic causes, 122 individuals were affected by HI, and 205 were unaffected. The male/female ratio of those enrolled for genetic investigations was 0.79 (*n* = 145/182). The mean age of diagnosis of HI was 4.3 ± 2.6 years. Most cases (89.36%, *n* = 109/122) reviewed were prelingual. Pedigree analysis suggested autosomal recessive inheritance in 46.3% (*n* = 38/82) of families. Most HI participants from familial cases had nonsyndromic HI (94.2%, *n* = 115/122). Waardenburg syndrome was found in three participants out of seven participants who presented with syndromic HI, while three other participants manifested signs of Usher syndrome and one with suspected Noonan syndrome. Molecular analysis did not find pathogenic variants in *GJB2* or *GJB6*-D3S1830 deletion in any of the probands tested (*n* = 27/122; 22.13%) or 100 non-affected control participants. **Conclusions:** This study revealed an overall late diagnosis (mean at 4.3 years) of HI in Rwanda. Most cases were of unknown origin or putative environmental origin (76.4%), with meningitis predominating as the acquired cause of early-onset HI. Among cases of putative genetic etiology, nonsyndromic HI accounted for the large majority (94.2%). However, *GJB2* and *GJB6* pathogenic variants were not identified in the screened proportion of the cohort. This study calls for the implementation of neonatal hearing screening as well as reinforcement of immunization programs to reduce the burden of acquired early-onset HI in Rwanda. Additional genomic studies, ideally using exome sequencing for familial cases, are needed.

## 1. Introduction

Congenital hearing impairment (HI) affects spoken language acquisition, and impacts affected individuals’ cognition, development, and social well-being [1]. Close to 5% (466 million) of people are estimated to have some threshold of disabling HI globally (WHO, 2023). About 1–2 in every 1000 live births are affected in high-income countries, whereas this prevalence is 6 times as high in low-income settings, and the burden is greatest in the Asian Pacific region, southern Asia, and sub-Saharan Africa [2]. In Rwanda, former reports estimated a 13.3% prevalence of HI among school children, with impacted wax (18%) and middle ear infections (5.8%) as the most common acquired factors implicated in cohorts investigated [3,4,5]. However, there is a dearth of data on the etiology and characterization of congenital and/or early-onset HI in Rwanda, and reported studies conducted among Rwanda’s population were limited to Kigali, the capital city.

Etiologically, HI can stem from genetic and/or acquired causes. Between 50 and 70% of the HI cases described are acquired, with meningitis being the most recurrent environmental cause in sub-Saharan Africa [6,7,8,9]. In general, about 30% of hereditary HI is associated with other organ disorders and/or clinical anomalies (syndromic HI). In 70% of the cases described as nonsyndromic, there is no additional clinical manifestation (HI as an isolated condition) [10]. Unlike syndromic HI, the early detection and diagnosis of nonsyndromic hearing impairment (NSHI) are not always possible in most infant hearing screenings, particularly in low- to middle-income countries [11].

NSHI is characterized by diverse genotypes with high allelic and locus heterogeneity [12]. Genetic variations in over 120 independent genes have been identified to cause hereditary NSHI [6]. Despite the high genetic and allelic heterogeneity, pathogenic variants in gap junction β-2 (*GJB2)* encoding connexin 26 protein and a large deletion in gap junction β-6 (*GJB6)* encoding connexin 30 protein were reported to contribute more than 70% of NSHI in populations of European and Asian descent [13,14]. Nevertheless, some recent studies have reported a lesser prevalence of 30–50% [10]. However, connexin mutation investigation in the sub-region showed no significant contribution in most African populations [6,15,16,17,18], with two notable exceptions in Ghanaian and Senegalese cohorts, among whom pathogenic variants in *GJB2* accounted for 42% and 34% of multiplex families segregating NSHI, respectively [19,20].

In Rwanda, the National Ear and Hearing Care Plan reports suspected hereditary causes in more than 50% of cases of HI among children; however, genetic investigations to establish the likely causal markers remain to be conducted [21]. In this study, using community-based nationwide recruitment, we characterized the clinical profile of HI in 348 families in Rwanda and studied the contribution of *GJB2* and *GJB6* variants. The data reveal the need to implement neonatal hearing screening services into the Rwandan primary healthcare system for early detection and diagnosis of childhood HI and to guide the provision of appropriate interventions.

## 2. Methods

### 2.1. Study Settings and Data Collection Tools

We used contact details of families with suspected cases of early-onset HI through different local collaborators (contact persons). Participants were recruited from seven inclusive schools (integrated regular schools with both disabled and non-disabled students), one hospital (Rwanda Military Hospital), and four independent villages/communities in Rwanda from September 2020 to October 2022. Our recruitment sites were in 12 districts out of 30 administrative districts in Rwanda (Figure 1). A well-designed structured questionnaire translated into Kinyarwanda which is the native spoken language of the participants, was used to collect data on socio-demographics, phenotype etiology, family history, and prenatal, neonatal, and postnatal clinical information of the affected individuals. Data collected were later recorded in the Research Electronic Data Capture (REDCap) [22].

### 2.2. Participant Recruitment

Our recruitment approach was phased into two stages. Stage one entailed contacting local collaborators, including head teachers at selected inclusive schools in Rwanda, heads of departments in hospitals that had ENT units, and the representatives of deaf associations in different regions across Rwanda. Some local collaborators were identified by the ENT specialist who was part of the study team and one of the co-authors of this manuscript. From his experience working with patients living with HI, the specialist knew various schools and facilities where hearing-impaired individuals were accommodated. The rest of the collaborators were identified through snowballing. The collaborators were invited and engaged in collaborator engagement sessions that explained the aim and objectives of the project to them. The collaborators initiated communication with the potential participants and provided their contacts. Family contacts received were followed up with phone calls, and parents or guardians were asked to respond to questions on the medical and family history of affected individuals. The study team explained the project to the potential participants via phone calls and community engagement project and facilitated the consenting process. Prospective participants (HI-affected families) were grouped into acquired/environmental cases and likely congenital/genetic cases. When acquired factors were suspected, i.e., HI was caused by infections, diseases, accidents, and other environmental factors, the recruitment process ended at stage one. On the other hand, when there was a strong suspicion of a likely genetic cause, the recruitment process continued to stage two. During stage two, further clinical data were collected by performing physical examinations and pure tone audiometry (PTA) using KUDUwave, a mobile audiometer (KUDUwave, Johannesburg, South Africa) for both affected participants and unaffected family relatives. Pre-existing audiological records for the affected participants were also reviewed if available. Air conduction thresholds were measured at 125 Hz, 250 Hz, 500 Hz, 1 kHz, 2 kHz, 4 kHz, and 8 kHz. Family pedigree data were obtained from at least the third generation (self-reported). Venous blood samples (4 mL) were collected from consenting families (proband, both parents, affected sibling/other relative, and unaffected sibling) by a phlebotomist. In simplex cases (only one affected individual in the entire family), blood samples were taken from the affected person, both parents, and an unaffected sibling if available. Whole blood samples were also collected from 100 healthy hearing controls randomly selected from University of Rwanda students.

### 2.3. Control Participants

In addition, we randomly recruited 100 ethno-lingually matched healthy hearing controls (students) from the University of Rwanda who reported neither self- nor family history of HI. These hearing control participants were aged 19 years and above.

### 2.4. Definitions

HI was defined as (1) acquired, if there was evidence of a relation between a self-reported environmental factor and the onset of the condition; (2) genetic, when more than one case was reported in the family, in case of consanguinity and clinically well-defined syndromic cases; (3) of unknown etiology, if there was no established likely cause, whether environmental or genetic in origin, as previously described by Wonkam et al. [23].

### 2.5. Molecular Analysis: GJB2 Sequencing and GJB6 Del (GJB6-D13S1830) Genotyping

DNA was extracted from venous blood samples collected from the study participants using a Maxwell RSC^®^ Whole Blood DNA Kit (Promega, Madison, WI, USA) following the manufacturer’s instructions in the laboratory of the College of Medicine and Health Sciences (CMHS), University of Rwanda, Kigali, Rwanda. At the Division of Human Genetics Laboratory, Department of Pathology, Faculty of Health Sciences, University of Cape Town, South Africa, isolated DNA concentration and purity were assessed using a Nano-drop ND-100 spectrophotometer (Nanodrop Technologies, Santa Clara, CA, USA) and a Qubit^®^ 3.0 fluorometer (Invitrogen, Kuala Lumpur, Malaysia), and agarose gel electrophoresis analysis was performed to check the integrity of the DNA.

Given the low prevalence of GJB2 and GJB6 mutations reported among populations of African ancestry, only probands identified from families with HI of putative genetic origin (*n* = 27/122; 22.13%) were selected to test for these markers, alongside 100 non-affected and apparently healthy control participants selected from the general Rwandan population of university students without a personal or family history of HI.

Previously reported primers for the *GJB2* gene were evaluated using BLAST^®^ (NIH, Bethesda, MD, USA) as previously recommended. Restricted fragment length polymorphism (RFLP) was performed using the restriction digest enzyme NciI (supplied by New England Biolabs Inc., Massachusetts, MA, USA, through Inqaba Biotec, Pretoria, South Africa) with the recognition site “CCSGG”. The coding region of the *GJB2* gene (exon 2) was amplified and resolved on 2% agarose gel for 1.5 h and viewed under UV light in the Division of Human Genetics, University of Cape Town, South Africa. We investigated del (*GJB6*-D13S1830) using the method and primers described by del Castillo et al., 2002 [14,24]. Details of the primer pair sequence and the reaction conditions are in Appendix A.

### 2.6. Data Analysis

Descriptive statistics and non-parametric statistical tests were performed for all comparisons using R software version 4.3.1 for all computations (IBM Corp., Armonk, NY, USA). The degree of HI was classified into profound, severe, moderate, and mild with corresponding types (sensorineural, conductive, and mixed HI) according to the American Speech-Language-Hearing Association (ASHA) classification [25]. Family pedigrees were analyzed following the Mendelian laws to define the likely pattern of inheritance.

## 3. Results

### 3.1. The Entire Cohort

A total of 422 participants were enrolled in this study (Table 1), with a male/female ratio of 1.03 (215/207). The majority (*n* = 216/422; 51.23%) denied any known acquired cause of HI. All participants were black Africans, mostly agedbetween 11 and 25 years (*n* = 308/422; 72.98%), with a primary level of formal education for most participants (*n* = 254/422, 60.18%).

We received 616 contacts from families with suspected early-onset HI cases. We recruited 348 families, comprising 422 affected individuals (including 122 affected identified from 82 families for genetic testing).

The male/female ratio of those enrolled for genetic testing was 0.79 (145/182). Most of the affected participants (52.48%, *n* = 64/122) reported no medical diagnosis performed before the time of recruitment in this study. The mean age at diagnosis was 4.3 ± 2.6 years (*n* = 58/122, 47.52%) among those who self-reported to have received a diagnosis from a healthcare provider by the time of recruitment.

A detailed diagram showing the flow and outcome of the recruitment process is described in Figure 2.

### 3.2. Families and Individuals with Putative Genetic Origin and Pedigree Analysis

Based on clinical evaluation, a total of 82/348 families (23.5%) were suspected to have HI of putative genetic origin. Within these 82 families with putative genetic origin, a physical medical auditory exam was performed for a total of 327 consenting participants, and blood samples were collected for genetic investigations. Among the 327 participants, 122 were affected (82 probands and 40 affected relatives) and 205 relatives were unaffected. Most of the affected participants (*n* = 120/122, 98.36%) were siblings. These were all the individuals whose clinical history (with the aid of a detailed questionnaire) excluded any environmental cause of HI. Ninety-four individuals with putative genetic HI were not enrolled for genetic testing because of financial difficulties and work that hindered parents or siblings at home from reaching the recruitment sites (schools and specific community locations agreed upon with our collaborators). Furthermore, three multiplex families declined our invitation to participate in genetic testing despite having verbally consented. Among the 82 families with putative genetic origin, 38 were multiplex families with at least two individuals affected by HI, and 44 were simplex, including only one individual affected by HI. At least one individual from 27 families segregating putative genetic HI was selected for genetic analysis of the *GJB2* and *GJB6* genes. Autosomal recessive was likely the mode of inheritance among 36 multiplex families, while autosomal dominant inheritance was likely in 2 multiplex families.

### 3.3. Nonsyndromic HI and Waardenburg, Usher, and Noonan Syndromes

Most of the cases from 82 families with putative genetic origin were mainly phenotypically nonsyndromic (94.27%, *n* = 115/122). A total of seven cases from five families clinically featured syndromic HI. Among them, three individuals from two different families presented with profound HI and features of Waardenburg syndrome type II, with no dystopia canthorum, and variable expression of HI not fully penetrant in the sibling from the family, as well as heterochromia, which was unilateral or bilateral within one family, and skin discoloration which was not penetrant in 2/3 of individuals in one family (Figure 3A,B,E).

Three other participants from two families showed signs of Usher syndrome associated with hearing impairment, and in two cases retinitis pigmentosa was observed (Figure 3F). In addition, another participant presented syndromic HI with features of Noonan syndrome (Appendix A).

### 3.4. Age of Onset and Diagnosis of HI

Hearing impairment in children was classified in relation to the time of onset as prelingual (before 2 years), perilingual (between 2 and 4 years), and postlingual (after 4 years), as described by Wonkam et al. [23]. Most of the affected individuals were reported by parents/guardians to have manifested signs of HI before 2 years of age, while postlingual onset was the least observed in our cohort (Figure 1C,D).

### 3.5. Other Putative Non-Genetic Etiologies of HI

Unknown congenital origin was the most common (51.23%; *n* = 216/422) cause reported, followed by meningitis (21.18%, *n* = 89/422), undetermined diseases (11.88%; *n* = 50/422), and birth asphyxia (7.38%, *n* = 31/422). Other causes that were reported included low birth weight (0.49%, *n* = 2/422), prematurity (1.47%; *n* = 6/422), head trauma (3.43%; *n* = 14/422), and other medical conditions (2.94%; *n* = 12/422) like complicated malaria, epilepsy, mumps, ear atresia, and otitis (Table 2).

### 3.6. Audiological Phenotypes

Pure tone audiometry was performed for 44 affected probands, 38 affected relatives, and 118 putatively unaffected relatives who were able to follow instructions. Eighty-two (41%, *n* = 82/200) participants who underwent audiological testing had different degrees of HI (Table 3). The audiological evaluation results mainly showed profound HI of the sensorineural type. Mixed hearing loss was the least common type observed in our cohort (Table 3).

### 3.7. GJB2 and GJB6 Genotyping

None of the 27 probands screened or 100 control participants were positive for any pathogenic variant in *GJB2* or del (*GJB6*-D13S1830).

## 4. Discussion

To our knowledge, this is the most comprehensive and nationwide study for HI in Rwanda, and its outcomes will likely inform policy. Indeed, the study revealed an overall late diagnosis (mean at 4.3 years) of HI in Rwanda. Most cases were of unknown origin or putative genetic origin (51.23%), and meningitis predominated among the acquired causes of early-onset HI. This study calls for implementation of neonatal hearing screening as well as reinforcement of immunization programs to promote early diagnosis and reduce the burden of acquired early-onset HI in Rwanda. By investigating the most common HI genetic markers, *GJB*2 and *GJB6*, this study pioneers the molecular genetic investigations of early-onset HI in Rwanda. Like in most African populations, we did not identify pathogenic variants in *GJB2* or any *GJB6* targeted deletion del (*GJB6*-D13S1830) among the screened affected individuals and controls, thus showing that these frequently reported markers are not significant contributors to congenital HI in the Rwandan population. This observation calls for the need to identify other HI-causing genetic markers. This can be achieved by employing exome sequencing, particularly to investigate multiplex families.

In previous Rwandan studies, infections were identified as the primary cause of HI (64%) in the cohort investigated [26]. Acquired causes of HI like meningitis, birth asphyxia, and unspecified illnesses constituted the most common etiological factors in the HI cases reviewed. The current observations are consistent with earlier reports in sub-Saharan Africa, which implicated meningitis and other unknown congenital etiologies as the predominant causes of early-onset HI [8,19,23]. In Rwanda, similar to other sub-Saharan African countries located in what has been known as the meningitis belt extending from Senegal to Ethiopia, meningitis has claimed thousands of lives over the past couple of decades and caused multiple sequelae, including hearing loss [27,28]. Therefore, the strong correlation observed in this study between early-onset HI and meningitis can be accounted for by this historical context. This finding was expected given the known pathophysiology of meningitis that extends to the ear causing septic emboli that block the vascularization of the inner ear and leading to ischemia and eventually ossification of the cochlea [29].

While the majority (89.36%) of our participants reported prelingual onset of HI, only 36.06% were diagnosed between 1 and 5 years of age, and the majority (52.48%) had no medical diagnosis at the time of recruitment (Figure 1C,D). This may be attributed to the lack of neonatal screening programs and limited ENT services in Rwanda. Similarly, a mean age at medical diagnosis as high as 3.3 ± 1.2 years has been previously reported in Cameroon, which is very late compared to developed settings where the most delayed diagnosis of early-onset HI was reported in Canada to be 13.8 months [23]. Moreover, in developed countries where neonatal hearing screening is available, the rate of unknown causes of HI accounts for only 10% [30]. Poverty, lack of awareness of available medical services, and sometimes a delayed onset of symptoms, in which cases parents fail to notice progressive HI, are among the factors that may contribute to delayed or no diagnosis in Rwanda, as reported by Mukara et al. [26]. Furthermore, there is only one facility (Rwanda Military Hospital) where auditory brainstem response (ABR) testing is available to screen hearing status in neonates, infants, and children who are not suited for behavioral hearing testing or PTA. The associated cost, coupled to the fact that ABR services in Rwanda are not covered by community-based health insurance (CBHI), makes the service expensive for the average person (the majority of the affected families).

Most of the affected individuals (94.26%, *n* = 115/122) enrolled for genetic testing were phenotypically nonsyndromic. On the other hand, three out of seven affected individuals who presented with features of syndromic HI had clinical signs of Waardenburg syndrome type II with the characteristic variable expression of pigmentation defects like hypochromia, white forelock, and skin hypopigmentation of the limbs and no dystopia canthorum (Figure 3A,B,E) [8,9,19,31]. Usher syndrome is reported among 50% of all the people who present with deafness and retinopathies worldwide [32]. We performed OCT, fundus photography, and direct ophthalmoscopy for eight of nine participants who reported signs of Usher syndrome. In agreement with previous research, a significant percentage of those examined (37.5%, 3/8) showed typical signs like mild optic disc pallor, scattered peripheral bone-spicule-type pigment deposits, and narrowing of retinal arterioles consistent with retinitis pigmentosa (Figure 3F and Appendix A). The ninth participant was not cooperative for imaging.

Pathogenic variants in the *GJB2* coding region (exon 2) and *GJB6* deletion did not contribute to the condition in the cohorts investigated. This finding aligns our study with reports in Cameroon, Nigeria, and South Africa, where *GJB2* and *GJB6* had little to no contribution to nonsyndromic HI [15,31,33,34]. High rates of uncomplimentary marriages/inbreeding have been implicated in populations with high frequencies of deleterious *GJB2* alleles, linked to a founder effect [19,20]. Although the frequency of consanguineous unions in Rwanda has not been empirically examined, it can be inferred that consanguinity is not common due to legal prohibitions and the widespread social stigma and perception in Rwanda. The 2.4% consanguinity rate (Appendix A) recorded among first-degree cousins (two families) in this study further reflects the low prevalence of intra-family marriage in the Rwandan population.

Elucidating the underlying genetic markers and population-specific structural dynamics perpetuating the segregation of congenital NSHI in the cohort studied is imperative going forward. Comprehensive genetic investigations, particularly using exome sequencing, will generate data on the spectrum of genes and variants contributing to HI phenotype segregation and would refine the genotype–phenotype curation among families with NSHI in Rwanda.

While this study provides valuable insights into environmental and genetic etiologies of HI in Rwanda, certain limitations must be acknowledged. First, though guided by a detailed questionnaire, the causes of HI were self-reported; thus, the study is subject to respondent bias. Second, only 200 participants (*n* = 200/327, 61.16%) were cooperative for PTA performance, leaving a gap in the audiological assessment. Third, the study only evaluated two genetic markers, *GJB2* and *GJB6* genes, excluding other potential genetic markers that may contribute to HI. Future studies should ideally use whole exome and whole genome sequencing, particularly in multiplex families, to ensure a comprehensive investigation of genetic and allelic heterogeneity of HI genetics in Rwanda.

## 5. Conclusions

This study revealed an overall late diagnosis (mean at 3.4 years) of HI in Rwanda, and before this study, most of our participants had not received any form of clinical diagnosis. Most cases were of unknown origin or putative environmental or unknown origin (76.4%), with meningitis predominating among the acquired causes of early-onset HI. Among cases of putative genetic etiology, nonsyndromic HI accounted for the large majority (94.2%). However, *GJB2* and *GJB6* pathogenic variants were not identified. This study calls for the implementation of neonatal hearing screening as well as reinforcement of immunization programs to reduce the burden of acquired early-onset HI in Rwanda. Additional genomic studies ideally using exome sequencing for familial cases are needed.

## Figures and Tables

**Figure 1 genes-16-00257-f001:**
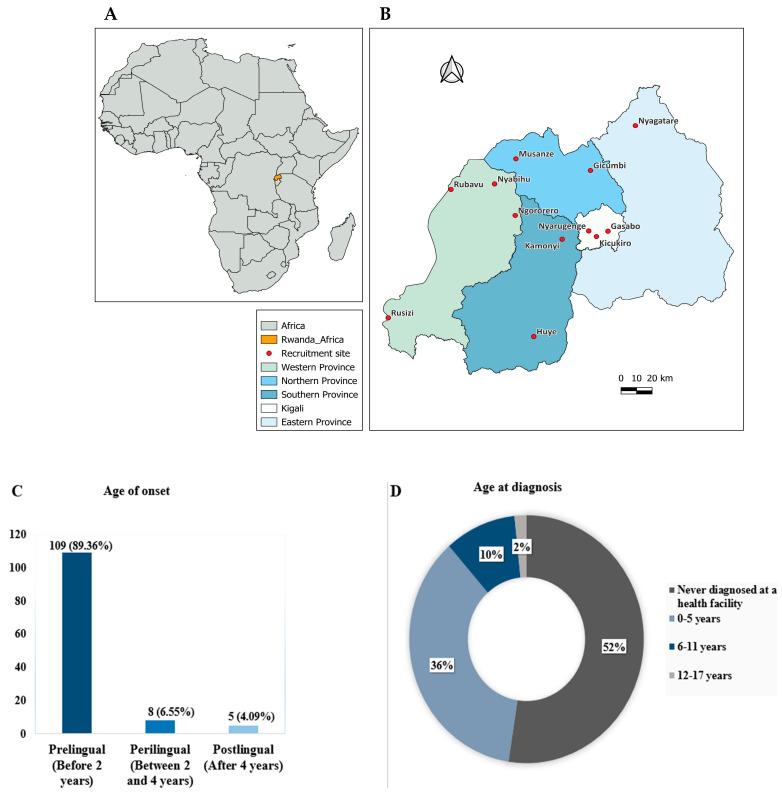
A map of Rwanda in Africa (**A**) depicting 12 nationwide recruitment sites (**B**). (**C**) Age of onset of HI. (**D**) Age at diagnosis.

**Figure 2 genes-16-00257-f002:**
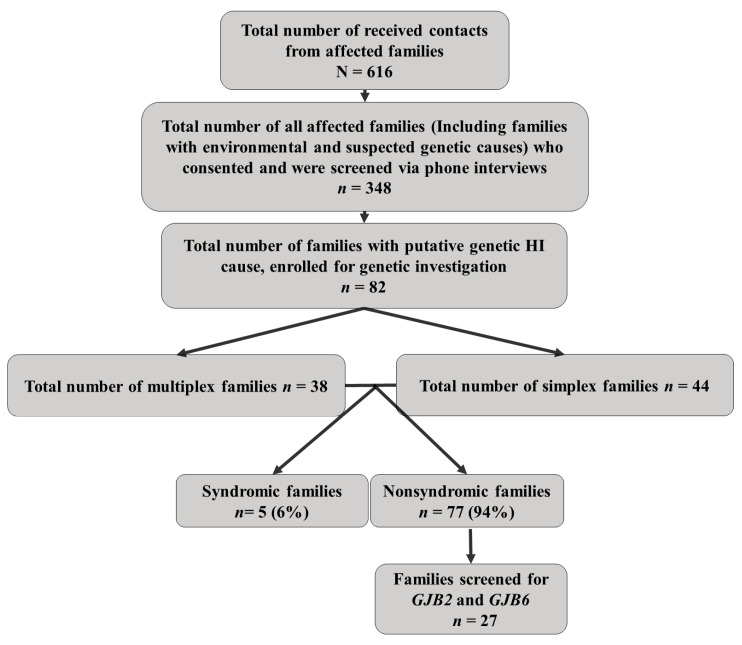
Diagram showing the flow and outcome of the recruitment process.

**Figure 3 genes-16-00257-f003:**
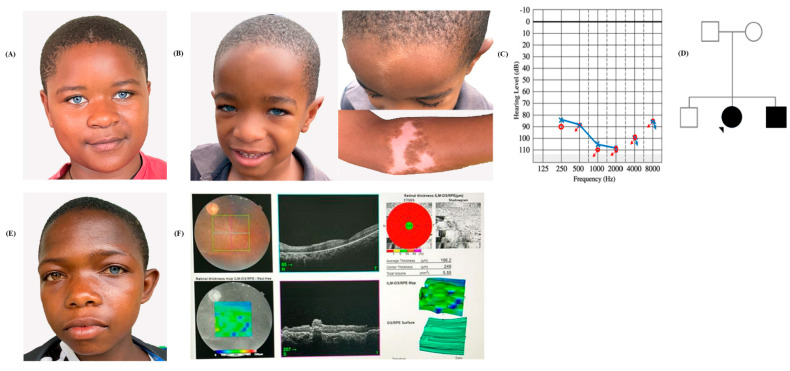
Clinical profiles of Waardenburg and Usher syndromes. (**A**,**B**). Familial case of a sister and her brother with bilateral profound HI, bilateral hypochromia, and pigmentary abnormalities of the hair and skin. (**C**) Audiogram of the female participant in (**A**); Red: Right ear; Blue: Left ear. and (**D**) the pedigree of her family–the arrow indicates proband. (**E**) Isolated case with bilateral profound HI, and unilateral hypochromic heterochromia, without dystopia canthorum. (**F**) Direct ophthalmoscopy and Ocular Coherence Tomography (OCT) image of a male participant from a multiplex family with Usher syndrome, demonstrating retinal pigment epithelial changes and loss of photoreceptors. The arrow indicates the proband.

**Table 1 genes-16-00257-t001:** Socio-demographic characteristics of the cohort.

Socio-Demographic Characteristics		*n*	%
Sex	Male	215	50.947
	Female	207	49.052
Age	<5	5	1.18
	5–10 yo	75	17.77
	11–25 yo	308	72.98
	>25	34	8.05
Level of formal education	No formal school/preschool	86	20.37
	Primary	254	60.18
	Secondary	65	15.4
	Tertiary	2	0.47
	Special	14	3.31
	Other	1	0.23
Ethnicity	Black African	422	100
HI etiology	Unknown	216	51.23
	Acquired	206	48.77

**Table 2 genes-16-00257-t002:** Etiologies of early-onset HI in the current study compared to some other African countries.

Country	Number of Patients	Meningitis	Birth Asphyxia	Low Birth Weight	Prematurity	Head Trauma	* Other Conditions	Unspecified Illnesses	Unknown Congenital Origin	*GJB2*- Mediated	Reference
Cameroon	582	34.4	-		0.9	0.3	-	-	32.6	0.34	[22]
Ghana	1104	3.9	-		0.5	1.5	10.8	6.3	53.8	27.2	[19]
Mali	117	40	-	15.7	-		25.7	-	11.3	Undetermined	[8]
Senegal	406	6.61	-		2.71	2.22	1.72	-	-	34.1	[9]
Rwanda (this study)	422	21.18	7.38	0.49	1.47	3.43	2.94	11.88	51.23	0	This study

* Epilepsy, complicated malaria, mumps, otitis, ear atresia.

**Table 3 genes-16-00257-t003:** Degree of HI and characterization.

Category of HI	Sensorineural HI	Conductive HI	Mixed HI	Total, *n* (%)
Profound	56	-	1	57 (69.5)
Severe	6	-	-	6 (7.3)
Moderate–severe	4	-	-	4 (4.9)
Moderate	1	9	1	11 (13.4)
Mild	4	-	-	4 (4.9)
Total, *n* (%)	71 (86.6)	9 (11)	2 (2.4)	82 (100)

## Data Availability

The original contributions presented in this study are included in the article/Appendix A. Further inquiries can be directed to the corresponding author.

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
