# Peer review of "Etiologies of Early-Onset Hearing Impairment in Rwanda"

_genes, 2025, doi:10.3390/genes16030257_

Round 1

Reviewer 1 Report

Comments and Suggestions for Authors

This study presents the etiologies of a large cohort of individuals with early-onset hearing impairment in Rwanda. Such work is crucial to broadening our knowledge of the etiologies of hearing loss to better represent that unique needs of a global populations. The authors used a community-based recruitment strategy to recruit a large cohort of individuals affected with early onset hearing impairment as was well as their unaffected relatives. They administered a survey and found that 21.18% were linked to meningitis, 51.23% had an unknown cause, and 23.6% had a putative genetic etiology. The authors then performed genetic testing of the genes GJB2 and GJB2 for a subset of these individuals and did not identify and variants in GJB2 or GJB6deletions.

This is potentially the first and certainly the largest study of the etiology of early-onset hearing impairment in Rwanda. The authors demonstrate that the prevalence of syndromic hearing loss in this population is low compared to other study populations, and GJB2 does not appear to be a substantial contributing factor to hearing loss in this population. The authors demonstrate a robust recruitment strategy and present novel results in the Rwandan population. However, this work would benefit from streamlining of how results are presented, and careful attention should be paid to whether it is probands vs. relatives being described. The authors should also be careful not to overstate their findings or conclusions on the genetic etiologies of hearing loss in this cohort, as a limited number of the probands had GJB2 testing.

Major points:

  1. Abstract, line 32 and line 41-43: It is misleading to the reader to state that 82 families were identified for genetic testing, when based on the following results, it appears that only 27 families underwent genetic testing for GJB2/GJB6. It could also be misleading to state that there are no GJB2/GJB6variants identified in the “entire cohort” when it was only 27 affected individuals that had that testing out of a cohort of 348 families.
  2. Methods: The use of community-based nationwide recruitment, such as forming relationships with local collaborators, teachers, and leaders of local deaf organizations, is an important way to reach this population and something the readers may want to hear more about. How were the local collaborators identified and how were these relationships forged? How did participants consent to take place in the research and who facilitated the informed consent process?
  3. The authors should include a table summarizing the cohort demographics and clinical characteristics, focusing on affected probands in the family. It sounds like an extensive survey and medical record review was conducted, and it would be helpful to be able to see (easily summarized as a table) some of these data. Furthermore, a supplementary table with participant-level clinical data should be included.
  4. Please consider re-phrase your results throughout to focus on testing performed for probands (rather than unaffected family members) and frame it that way for your writing. For example, lines 265-267, should read, “pure tone audiometry was performed for ### affected probands, ### affected relatives, and ### putatively unaffected relatives.” (It would be interesting to know if any individuals with reportedly normal hearing we found to have HI. ) This type of edit could help clarify section 3.3 as well, referring to proband and affected relatives vs. total number of affected individuals. Were the affected relatives siblings or parents? It was challenging at points throughout the manuscript to distinguish total individuals vs. probands vs. relatives and I would suggest reviewing this for clarity. When calculating the proportion of syndromic vs. nonsyndromic and other similar proportions throughout the manuscript, this should be based on the number of probands (one from each family), not total affected.
  5. Your methods detail the molecular techniques used to isolate DNA and sequence GJB2 and looks for GJB6 deletions, but do not describe how sequencing data was analyzed to assess for variants. What variant classification guidelines were used? Could variants of uncertain significance be identified using this methodology? Results section 3.3 says GJB2 genotyping was performed for 27 proband; was this just genotyping or was the whole sequencing for GJB2 exon 2 obtained? How were the 27 probands in particular identified selected for this testing?
  6. Described methods includes use of control participants from the University, but the manuscript does comment on how these controls and information was used in the downstream analysis.
  7. The discussion section lacks comment on the limitations on this study.

Minor points:

  • Line 60: rephrase writing for clarity, “rate increases to 6-fold in low-income settings.”
  • Lines 85-87: More recent literature of clinically and ancestrally diverse cohorts shows a lower prevalence of GJB2-assocaited hearing loss that accounts for ~30% of hereditary hearing loss cases. You may consider adding more recent citations here.
  • Line 212 states that 327 participants were audiologically examined, but line 265, only 200 individuals had pure tone audiometry. What sort of testing did the others have? How did you decide which individuals would undergo PTA?
  • Line 263: Please include genetic etiology, GJB2-mediated, or suspected genetic etiology to the table, if possible. Perhaps “unknown congenital origin” is a stand-in for genetic/hereditary, but this could be make more clear. Including numbers (n) as well as percent across the table could help readers understand the scale to which similar research has been performed in other African countries.
  • Line 323 states that 122 underwent genetic testing, but results section (line 274) states only 27 probands had GJB2/GJB6 analysis. Would “genetic evaluation” be more appropriate to describe the situation on line 232? Also, line 232 states 118/122 who had genetic evaluation had a nonsyndromic presentation, but that there were 7 affected individuals with a syndromic presentation. This seems to be a numerical discrepancy somewhere. Should these read 115/122, or it does on line 225?
  • Lines 331-333: These seems like numbers that should be included in the results section, as it is not mentioned anywhere else that 9 individuals had HI with progressive vision loss
  • Line 345: 2.4% consanguinity is reported in the discussion section, but I could not find this information anywhere above. Please see major point on adding a demographics and clinical characteristics table in the results.

Author Response

Reviewer 1

Major points:

  1. Abstract, line 32 and line 41-43: It is misleading to the reader to state that 82 families were identified for genetic testing, when based on the following results, it appears that only 27 families underwent genetic testing for GJB2/GJB6. It could also be misleading to state that there are no GJB2/GJB6 variants identified in the “entire cohort” when it was only 27 affected individuals that had that testing out of a cohort of 348 families.

Response:  Thank you for this important observation. We have revised and rephrased this statement into a finding-based one. Line 41 – 43 now reads as follows:

Molecular analysis did not find pathogenic variant in GJB2 nor the GJB6-D3S1830 deletion in none of the probands tested (n = 27/122; 22.13%) and 100 non-affected control participants.

  1. Methods: The use of community-based nationwide recruitment, such as forming relationships with local collaborators, teachers, and leaders of local deaf organizations, is an important way to reach this population and something the readers may want to hear more about. How were the local collaborators identified and how were these relationships forged? How did participants consent to take place in the research and who facilitated the informed consent process?

Response: Thank you for your valuable questions about the recruitment process. We have revised the section and detailed the recruitment process in line 121 – 131, line 133 - 135 as follows:

Some local collaborators were identified by the ENT specialist who was part of the study team and one of the co-authors of this manuscript. From his experience working with patients living with HI, the specialist knew various schools and facilities where hearing-impaired individuals were facilitated and accommodated. The rest of the collaborators were identified through snowballing. The collaborators were invited and engaged in collaborator engagement sessions that explained the aim and objectives of the project to them. The collaborators initiated communication with the potential participants and provided their contacts.

….

The study team explained the project to the potential participants via phone calls and community engagement project and facilitated the consenting process.

  1. The authors should include a table summarizing the cohort demographics and clinical characteristics, focusing on affected probands in the family. It sounds like an extensive survey and medical record review was conducted, and it would be helpful to be able to see (easily summarized as a table) some of these data. Furthermore, a supplementary table with participant-level clinical data should be included.

Response: Thank you for your valuable feedback regarding the characteristics of the cohort presented. A supplementary table of summarized clinical data is now provided (Table S2). In addition, we have included a table summarizing Socio-demographic characteristics (Table 1), with the following description in the text (lines 204-208):

A total of 422 participants were enrolled in this study with a male-to-female ratio of 1.03 (215/207). The majority (n = 216/422; 51.23%) denied any known acquired cause of HI. All participants were black Africans, mostly aged between 11 and 25 years (n = 308/422; 72.98), with primary level of formal education for most participants (n = 254/422, 60.18%).

  1. Please consider re-phrase your results throughout to focus on testing performed for probands (rather than unaffected family members) and frame it that way for your writing. For example, lines 265-267, should read, “pure tone audiometry was performed for ### affected probands, ### affected relatives, and ### putatively unaffected relatives.” (It would be interesting to know if any individuals with reportedly normal hearing we found to have HI). This type of edit could help clarify section 3.3 as well, referring to proband and affected relatives vs. total number of affected individuals. Were the affected relatives siblings or parents? It was challenging at points throughout the manuscript to distinguish total individuals vs. probands vs. relatives and I would suggest reviewing this for clarity. When calculating the proportion of syndromic vs. nonsyndromic and other similar proportions throughout the manuscript, this should be based on the number of probands (one from each family), not total affected.

Response: Thank you for your valuable and constructive suggestions. We have revised the results sections, and the following changes appear in lines 228 – 230; 230 – 231; 295 – 286;

Among the 327 participants 122 were affected (82 probands and 40 affected relatives) and 205 unaffected relatives.

Most of the affected participants (n= 120/122, 98.36%) were siblings.

Pure tone audiometry was performed for 44 affected probands, 38 affected relatives, and 118 putatively unaffected relatives who were able to follow instructions.

  1. Your methods detail the molecular techniques used to isolate DNA and sequence GJB2 and looks for GJB6 deletions, but do not describe how sequencing data was analyzed to assess for variants. What variant classification guidelines were used? Could variants of uncertain significance be identified using this methodology? Results section 3.3 says GJB2 genotyping was performed for 27 proband; was this just genotyping or was the whole sequencing for GJB2 exon 2 obtained? How were the 27 probands in particular identified selected for this testing?

Response: Thank you for these important questions. We have revised the method section and made the necessary changes in lines 187 – 196 as follows:

Restricted fragment length polymorphism (RFLP) was performed using the restriction digest enzyme NciI (supplied by New England Biolabs Inc., Massachusetts, MA, USA, through Inqaba Biotec, Pretoria, South Africa) with the recognition site “CCSGG”. The coding region of the GJB2 gene (exon 2) was amplified and resolved on 2% agarose gel for 1.5 h and viewed under UV light in the Division of Human Genetics, University of Cape Town, South Africa. We investigated del (GJB6-D13S1830) using the method and primers described by del Castillo et al., 2002. Details of primer pairs sequence, and reaction conditions are in Supplementary Table 1.

  1. Described methods includes use of control participants from the University, but the manuscript does comment on how these controls and information was used in the downstream analysis.

Response: Thank you for your important observation. We have revised the manuscript, and changes were made as far as the control participants are concerned; lines 180-186; 296 – 297 read as follows:

Given the low prevalence of GJB2 and GJB6 mutations reported among populations of African ancestry, only proband identified from families with HI of putative genetics origin (n = 27/122; 22.13%) was selected to test for these markers, alongside with 100 non-affected and apparently healthy control participants selected from the general Rwandan population, without a personal and family history of HI.

None of the 27 probands screened or 100 control participants were positive for any pathogenic variant in GJB2 or del (GJB6-D13S1830).

  1. The discussion section lacks comment on the limitations on this study.

Response: We thank you or your comment. We have revised the manuscript and include a limitation section appearing in lines 375 – 383 as follows:

While this study provides valuable insights into environmental and genetic etiologies of HI in Rwanda, certain limitations must be acknowledged. First, though guided by a detailed questionnaire, the causes of HI were self-reported thus the study is subject to respondent bias. Second, only 200 participants (n = 200/327, 61.16%) were cooperative for PTA performance leaving a gap in the audiological assessment. Third, the study only evaluated two genetic markers, GJB2 and GJB6 genes, excluding other potential genetic markers that may contribute to HI. Future studies should use ideally whole exome and whole genome sequencing, particularly in multiplex families, to ensure a comprehensive investigation of genetic and allelic heterogeneity of HI genetics in Rwanda.

Minor points:

ï‚· Line 60: rephrase writing for clarity, “rate increases to 6-fold in low-income settings.”

Response: Thank you for your comments. We have made the following changes to clarify the information as appears in lines 59 – 62:

About 1-2 in every 1000 live births are affected in high-income countries, whereas this prevalence is 6 times in low-income settings, and the burden is greatest in the Asian Pacific region, southern Asia, and sub-Saharan Africa.

ï‚· Lines 85-87: More recent literature of clinically and ancestrally diverse cohorts shows a lower prevalence of GJB2-assocaited hearing loss that accounts for ~30% of hereditary hearing loss cases. You may consider adding more recent citations here.

Response: Thank you for your suggestion. We have revised and added more recent reference appearing in lines 84 – 85 as follows:

Nevertheless, some recent studies have reported a lesser prevalence of 30-50% [10].

ï‚· Line 212 states that 327 participants were audiologically examined, but line 265, only 200 individuals had pure tone audiometry. What sort of testing did the others have? How did you decide which individuals would undergo PTA?

Response: Thank you for your questions. We have revised these statements to establish clarity and the changes made appear in lines 233 – 234; 295 – 296 as follows:

A physical medical auditory exam was performed for a total of 327 consenting participants.

Pure tone audiometry was performed for 44 affected probands, 38 affected relatives, and 118 putatively unaffected relatives who were able to follow instructions.

ï‚· Line 263: Please include genetic etiology, GJB2-mediated, or suspected genetic etiology to the table, if possible. Perhaps “unknown congenital origin” is a stand-in for genetic/hereditary, but this could be make more clear. Including numbers (n) as well as percent across the table could help readers understand the scale to which similar research has been performed in other African countries.

Response: Thank you for your valuable suggestion. We have revised and added a column that demonstrates the GJB2 contribution in the presented studies, line 292 (Table 2)

ï‚· Line 323 states that 122 underwent genetic testing, but results section (line 274) states only 27 probands had GJB2/GJB6 analysis. Would “genetic evaluation” be more appropriate to describe the situation on line 232? Also, line 232 states 118/122 who had genetic evaluation had a nonsyndromic presentation, but that there were 7 affected individuals with a syndromic presentation. This seems to be a numerical discrepancy somewhere. Should these read 115/122, or it does on line 225?

Response: Thank you for your valuable observation. We have revised the statement and corrected the numerical error. The corrected statement appears in lines 347 – 348 as follows:

Most of the affected individuals (94.26%, n = 115/122) enrolled for genetic testing were phenotypically non-syndromic.

ï‚· Lines 331-333: These seems like numbers that should be included in the results section, as it is not mentioned anywhere else that 9 individuals had HI with progressive vision loss

Response: Thank you for this point raised. Kindly note that the progressive vision loss was self-reported and upon investigation with OCT, fundus photography, and direct ophthalmoscopy, only 3 were confirmed. These 3 individuals were accounted for among the participants with syndromic HI.

ï‚· Line 345: 2.4% consanguinity is reported in the discussion section, but I could not find this information anywhere above. Please see major point on adding a demographics and clinical characteristics table in the results.

Response: Thank you for this important observation. We have included a supplementary table (Table S2 is now referred in the text, line 267) which summarized the clinical characteristics of the cohort and includes the consanguinity data.

Reviewer 2 Report

Comments and Suggestions for Authors

The manuscript by Uwibambe et al. analyzed the etiologies of early-onset hearing impairment (HI) in 12 of 30 districts in Rwanda. The authors enrolled 348 families comprising 422 individuals with HI and investigated the origins of their HI. 51% of cases had unknown cases for HI, while 49% were due to acquired causes, mostly meninigitis (21%). In addition, the authors show that the vast majority of patients had no medical diagnosis at the time of recruitment despite a staggering 89% of cases reporting prelingual onset for their HI. The authors call for the implementation of universal newborn hearing screening in Rwanda to direct interventions, as well as for the need to control meningitis as an acquired cause of HI.

As regards genetic causes for HI, the authors diagnosed Waardenburg type II syndrome in 3/7 syndromic cases of HI, 3/7 cases of Usher syndrome and 1/7 cases of Noonan syndrome. No genetic causes for nonsyndromic hearing impairment were identified, despite performing sequencing the coding exon of GJB2 and testing for the most common deletion at the DFNB1 locus in 27 probands with putative genetic origin of their HI.

The article is clearly written and conclusions and calls for action clearly follow from the data. However, I have some points for clarification.

MAJOR POINTS

(1) Which criteria were used to classify families as having a likely genetic origin for their HI? This is not explained and it is an important point that must be clarified. Why did the remainder of individuals with HI of unknown origin not fit the criteria for a genetic origin? Given that a category of "other unspecified diseases" (12%) was included among acquired causes, this leaves 216 - 122 = 94 individuals with unknown cause for their HI unaccounted for. Could the authors explain their criteria and suggest putative causes for the HI of these 94 patients? Maybe it is that thera are not enough data to classify those individuals, but if that is the case, it should be indicated.

(2) Which were the criteria to select the 27 probands that were screened for GJB2 variants and DFNB1 deletions? Were they audiologically characterized? Again, this should be indicated.

MINOR POINTS

(3) Informed consent and ethical committee informations on the study are missing from the manuscript. This important oversight should be corrected.

(4) Methods, lines 175-176 "PCR results were validated by Sanger sequencing of 10% of samples" What do the authors mean? Do they refer to the multiplex PCR for del (GJB6-D13S1830) screening? If so, how can sequencing verify this negative result? If they refer to the amplification of the complete coding region of GJB6, why perform the sequencing? Please explain.

(5) Introduction, line 73. I believe that the authors should rephrase it as "there is no additional clinical manifestation".

(6) Introduction, lines 73-77. I do not understand this sentence. When infant hearing screenings are performed, it is difficult to miss the diagnosis. Please clarify.

Comments on the Quality of English Language

The authors should revise their manuscript to remove typos, and interrupted sentences (e.g. line 279).

Author Response

Reviewer 2

MAJOR POINTS

  • Which criteria were used to classify families as having a likely genetic origin for their HI? This is not explained, and it is an important point that must be clarified. Why did the remainder of individuals with HI of unknown origin not fit the criteria for a genetic origin? Given that a category of "other unspecified diseases" (12%) was included among acquired causes, this leaves 216 - 122 = 94 individuals with unknown cause for their HI unaccounted for. Could the authors explain their criteria and suggest putative causes for the HI of these 94 patients? Maybe it is that thera are not enough data to classify those individuals, but if that is the case, it should be indicated.

Response: Thank you for this critical observation about the criteria followed to conclude on the cause as putative genetic. We have revised the section and included the criteria as appears in lines 235 – 239 as follows:

Among the 327 participants 122 were affected (82 probands and 40 affected relatives) and 205 unaffected relatives. Most of the affected participants (n= 120/122, 98.36%) were siblings. These were all the individuals whose clinical history (with the aid of a detailed questionnaire) excluded any environmental cause of HI.

As for affected individuals with putative genetic causes that were not accounted for. We have added a clarification point that appears in lines 240 – 244 and reads as follows:

Ninety-four individuals with unknown cause for their HI were not enrolled for genetic testing because of financial difficulties and work that hindered parents or siblings at home from reaching the recruitment sites (schools and specific community locations agreed upon with our collaborators). Furthermore, three multiplex families declined our invitation to participate in genetic testing despite having verbally consented.

(2) Which were the criteria to select the 27 probands that were screened for GJB2 variants and DFNB1 deletions? Were they audiologically characterized? Again, this should be indicated.

Response: Thank you for this important question. We have indicated the criteria for testing GJB2 and GJB6 in line 180 – 185 which read as follows:

Given the low prevalence of GJB2 and GJB6 mutations reported among populations of African ancestry, only proband identified from families with HI of putative genetics origin (n = 27/122; 22.13%) was selected to test for these markers, alongside with 100 non-affected and apparently healthy control participants selected from the general Rwandan population of university student, without a personal and family history of HI.

MINOR POINTS

(3) Informed consent and ethical committee informations on the study are missing from the manuscript. This important oversight should be corrected.

Response: Thank you for this comment. The informed consent and ethical committee information appear in lines 411 –417 and reads as follows:

Institutional Review Board Statement: The study was conducted in accordance with the Declaration of Helsinki, and approved by the Institutional Review Board of the College of Medicine and Health Sciences – University of Rwanda (N0 278/CMHS IRB 2020), and the University of Cape Town’s Faculty of Health Sciences’ Human Research Ethics Committee (HREC 039/2024).

Informed Consent: Written and informed consent was obtained from participants aged 18 years or older, to participate in this study and for publication. For minors (individuals younger than 18 years), they assented, and written informed consent was obtained from their parents/guardians.

(4) Methods, lines 175-176 "PCR results were validated by Sanger sequencing of 10% of samples" What do the authors mean? Do they refer to the multiplex PCR for del (GJB6-D13S1830) screening? If so, how can sequencing verify this negative result? If they refer to the amplification of the complete coding region of GJB6, why perform the sequencing? Please explain.

Response: Thank you for these important questions. Please note that there has been a mistake mentioning Sanger sequencing. We have revised the method section and made the necessary corrections in lines 171 – 179 as follows:

Restricted fragment length polymorphism (RFLP) was performed using the restriction digest enzyme NciI (supplied by New England Biolabs Inc., Massachusetts, MA, USA, through Inqaba Biotec, Pretoria, South Africa) with the recognition site “CCSGG”. The coding region of the GJB2 gene (exon 2) was amplified and resolved on 2% agarose gel for 1.5 h and viewed under UV light in the Division of Human Genetics, University of Cape Town, South Africa. We investigated del (GJB6-D13S1830) using the method and primers described by del Castillo et al., 2002. Details of primer pairs sequence, and reaction conditions are in Supplementary Table 1.

(5) Introduction, line 73. I believe that the authors should rephrase it as "there is no additional clinical manifestation".

Response: Thank you for this point of correction. We have addressed the mistake and now line 74 reads as follows: There is no additional clinical manifestation (HI as an isolated condition).

(6) Introduction, lines 73-77. I do not understand this sentence. When infant hearing screenings are performed, it is difficult to miss the diagnosis. Please clarify.

Response: Thank you for your response. The sentence has been revised to read as follows in lines 75 – 77:

Unlike syndromic HI, the early detection and diagnosis of non-syndromic hearing impairment (NSHI) are not always detectable in most infant hearing screenings, particularly in low to middle-income countries [11].

Reviewer 3 Report

Comments and Suggestions for Authors

row 142            replace ”...selected University of Rwanda students” with ” selected from University of Rwanda students”

row 215            for the syndromic cases of HI missing molecular genetic analyses that are essential for genetic counselling and for a differential diagnosis; in absence of these molecular investigations the presentation of these cases it's a bit out of context; try to present these cases in other way or to give up the presentation of them.

row 255            table 1 is irrelevant because for the other studies that concern HI in Africa, summing up the percentages does not give the results of 100%

row 272-277      The sentences: ”Most case were of unknown origin or putative environmental origin (76.4%), with meningitis predominating the acquired cause of early on set HI. This study calls for implementation of neonatal hearing screening as well as reinforcement of immunization programs to reduce the burden of acquired early-onset HI in Rwanda. Most potential cause of HI (51.23%) was unknown, and a good proportion of these cases could be suspected to be genetic.” are not clear. It requests a reformulation.

row 279            The sentence: ”Like in most African populations.” is a nonsense sentence.

rows 279-282    Reformulate the sentence: ”We did not identify pathogenic variants in GJB2 and any GJB6 targeted deletion del (GJB6-D13S1830), thus, showing that these frequently reported markers are not contributors to congenital HI in the Rwandan population.” It is not clear.

rows 282-284    Reformulate the sentence: ” This observation illuminates the need to identify other HI genetic markers, likely using exome sequencing particularly in multiplex families which segregate HI in Rwanda.”

row 287               replace ”... common factors in HI cases..” with ”... common etiological factors in HI cases..”

rows 225-227 and rows 319-326 Try to correlate the information between these sentences placed in different part of paper: ”Three other participants from 2 families showed signs of Usher syndrome associated with hearing impairment and in 2 cases retinitis pigmentosa was observed (Figure 3F).” and ” Usher syndrome is reported among 50% of all the people who present with deafness and retinopathies worldwide [32]. We performed OCT, fundus photography, and direct ophthalmoscopy for 8 of 9 participants who were affected with HI and reported progressive visual loss. In agreement with previous research, a significant percentage of those examined (37.5%, 3/8) showed typical signs like mild optic disc pallor, scattered peripheral bone-spicule type pigments deposits, and narrowing of retinal arterioles consistent with retinitis pigmentosa (Figure 3F and Supplementary Figure S2). The 9th participant was not cooperative for imaging.”

Author Response

Reviewer 3

row 142 replace ”...selected University of Rwanda students” with ” selected from University of Rwanda students”

Response: Thank you for your feedback. We have incorporated the suggested changes in lines 153 – 155 as follows:

Whole blood samples were also collected from 100 healthy hearing controls randomly selected from University of Rwanda students.

row 215 for the syndromic cases of HI missing molecular genetic analyses that are essential for genetic counselling and for a differential diagnosis; in absence of these molecular investigations the presentation of these cases it's a bit out of context; try to present these cases in other way or to give up the presentation of them.

Response: Thank you for this point raised. Kindly note that the participants were considered for further molecular investigations, when possible, whose results will guide the genetics counselling later.

row 255 table 1 is irrelevant because for the other studies that concern HI in Africa, summing up the percentages does not give the results of 100%

Response: Thank you for your observation. Please note that the other studies were only weighed against factors they shared with the current study.  Please note that we have added, data the GJB2- mediated HI as requested by reviewer 1 (now Table 2, line 292).

row 272-277 The sentences:” Most cases were of unknown origin or putative environmental origin (76.4%), with meningitis predominating the acquired cause of early on set HI. This study calls for implementation of neonatal hearing screening as well as reinforcement of immunization programs to reduce the burden of acquired early-onset HI in Rwanda. Most potential cause of HI (51.23%) was unknown, and a good proportion of these cases could be suspected to be genetic.” are not clear. It requests a reformulation.

Response: Thank you for your comment. We have revised the statements and changes appear in lines 308 – 312 as follows:

Most cases were of unknown origin or putative genetic origin (51.23%) and meningitis predominated the acquired causes of early onset HI. This study calls for implementation of neonatal hearing screening as well as reinforcement of immunization programs to promote early diagnosis and reduce the burden of acquired early-onset HI in Rwanda.

row 279 The sentence: ”Like in most African populations.” is a nonsense sentence.

Response: Thank you for this important observation. We have revised the sentence and corrected the grammatical error. A corrected statement reads as follows in lines 313 – 317:

Like in most African populations, we did not identify pathogenic variants in GJB2 and any GJB6 targeted deletion del (GJB6-D13S1830) among the screened affected individuals and controls, thus, showing that these frequently reported markers are not significant contributors to congenital HI in the Rwandan population.

rows 279-282 Reformulate the sentence:  ”We did not identify pathogenic variants in GJB2 and any GJB6 targeted deletion del (GJB6-D13S1830), thus, showing that these frequently reported markers are not contributors to congenital HI in the Rwandan population.” It is not clear.

Response: Thank you for your feedback. We have revised the statement and reformulated it to read as follows in lines 314 – 317:

we did not identify pathogenic variants in GJB2 and any GJB6 targeted deletion del (GJB6-D13S1830) among the screened affected individuals and controls, thus, showing that these frequently reported markers are not significant contributors to congenital HI in the Rwandan population.

rows 282-284 Reformulate the sentence: ” This observation illuminates the need to identify other HI genetic markers, likely using exome sequencing particularly in multiplex families which segregate HI in Rwanda.”

Response: Thank you for your comment. We have revised the sentence to read as follows in lines 319 – 321

This observation calls for the need to identify other HI-causing genetic markers. This can be achieved by employing exome sequencing, particularly to investigate multiplex families.

row 287 replace ”... common factors in HI cases..” with ”... common etiological factors in HI cases..”

Response: Thank you for your suggestion. We have rephrased the sentence to read as follows in line 323:

The most common etiological factors in HI cases reviewed.

rows 225-227 and rows 319-326 Try to correlate the information between these sentences placed in different part of paper: ”Three other participants from 2 families showed signs of Usher syndrome associated with hearing impairment and in 2 cases retinitis pigmentosa was observed (Figure 3F).” and ” Usher syndrome is reported among 50% of all the people who present with deafness and retinopathies worldwide [32]. We performed OCT, fundus photography, and direct ophthalmoscopy for 8 of 9 participants who were affected with HI and reported progressive visual loss. In agreement with previous research, a significant percentage of those examined (37.5%, 3/8) showed typical signs like mild optic disc pallor, scattered peripheral bone-spicule type pigments deposits, and narrowing of retinal arterioles consistent with retinitis pigmentosa (Figure 3F and Supplementary Figure S2). The 9th participant was not cooperative for imaging.”

Response: Thank you for your comment. We have revised the statements to correlate. The changes were made to the second part as appears in lines 355 – 357 as follows:

Usher syndrome is reported among 50% of all the people who present with deafness and retinopathies worldwide [32]. We performed OCT, fundus photography, and direct ophthalmoscopy for 8 of 9 participants who reported signs of Usher syndrome.

Round 2

Reviewer 1 Report

Comments and Suggestions for Authors

I thank the authors for their improvements to the manuscript. I think it is now suitable for publication.

Comments on the Quality of English Language

One important typo in the abstract. Molecular analysis did not find pathogenic variant in GJB2 nor the GJB6-D3S1830 deletion in none of the probands tested (n = 27/122; 22.13%) and 100 non-affected control participants. Should be "in any of the probands tested"

I would recommend English Language proofreading for the manuscript to catch any other grammatical errors.

Reviewer 2 Report

Comments and Suggestions for Authors

I thank the authors by their courtesy in answering my queries. I have no more comments.